# Nicotinamide Prevents Apolipoprotein B-Containing Lipoprotein Oxidation, Inflammation and Atherosclerosis in Apolipoprotein E-Deficient Mice

**DOI:** 10.3390/antiox9111162

**Published:** 2020-11-21

**Authors:** Karen Alejandra Méndez-Lara, Nicole Letelier, Nuria Farre, Elena M. G. Diarte-Añazco, Núria Nieto-Nicolau, Elisabeth Rodríguez-Millán, David Santos, Victor Pallarès, Joan Carles Escolà-Gil, Tania Vázquez del Olmo, Enrique Lerma, Mercedes Camacho, Ricardo P Casaroli-Marano, Annabel F. Valledor, Francisco Blanco-Vaca, Josep Julve

**Affiliations:** 1Institut de Recerca de l’Hospital de la Santa Creu i Sant Pau i Institut d’Investigació Biomèdica de l’Hospital de la Santa Creu i Sant Pau (IIB-Sant Pau), 08041 Barcelona, Spain; kmendez1101@gmail.com (K.A.M.-L.); ediarte@santpau.cat (E.M.G.D.-A.); nurianieton@gmail.com (N.N.-N.); elisabethrodriguez78@gmail.com (E.R.-M.); mpallaresl@santpau.cat (V.P.); jescola@santpau.cat (J.C.E.-G.); mcamacho@santpau.cat (M.C.); rcasaroli@ub.edu (R.P.C.-M.); 2Departament de Bioquímica i Biologia Molecular, Universitat Autònoma de Barcelona, 08193 Barcelona, Spain; 3Departament de Biologia Cel·lular, Fisiologia i Immunologia. Facultat de Biologia, Universitat de Barcelona i Institut de Biomedicina de la Universitat de Barcelona (IBUB), 08028 Barcelona, Spain; nicole.letelier.t@gmail.com (N.L.); afernandezvalledor@ub.edu (A.F.V.); 4Servei de Bioquímica, Hospital de la Santa Creu i Sant Pau, IIB-Sant Pau, 08041 Barcelona, Spain; farre.nuria88@gmail.com; 5Barcelona Tissue Bank (BTB) & Donor Center, Banc de Sang i Teixits (BST), 08005 Barcelona, Spain; 6CIBER de Diabetes y Enfermedades Metabólicas Asociadas, CIBERDEM, 28029 Madrid, Spain; daymer11@hotmail.com; 7Servei d’Anatomia Patològica, Hospital de la Santa Creu i Sant Pau, IIB-Sant Pau, 08041 Barcelona, Spain; tvazquez@santpau.cat (T.V.d.O.); elerma@santpau.cat (E.L.); 8CIBER de Enfermedades Cardiovasculares (CIBERCV), 28029 Madrid, Spain; 9Departament de Cirurgia, Facultat de Medicina & Hospital Clínic de Barcelona, Universitat de Barcelona, 08036 Barcelona, Spain

**Keywords:** macrophage, cytokine, ATP-binding cassette (ABC) transporters, niacinamide, vitamin B3, cardiovascular disease

## Abstract

The potential of nicotinamide (NAM) to prevent atherosclerosis has not yet been examined. This study investigated the effect of NAM supplementation on the development of atherosclerosis in a mouse model of the disease. The development of aortic atherosclerosis was significantly reduced (NAM low dose: 45%; NAM high dose: 55%) in NAM-treated, apolipoprotein (Apo)E-deficient mice challenged with a Western diet for 4 weeks. NAM administration significantly increased (1.8-fold) the plasma concentration of proatherogenic ApoB-containing lipoproteins in NAM high-dose (HD)-treated mice compared with untreated mice. However, isolated ApoB-containing lipoproteins from NAM HD mice were less prone to oxidation than those of untreated mice. This result was consistent with the decreased (1.5-fold) concentration of oxidized low-density lipoproteins in this group. Immunohistochemical staining of aortas from NAM-treated mice showed significantly increased levels of IL-10 (NAM low-dose (LD): 1.3-fold; NAM HD: 1.2-fold), concomitant with a significant decrease in the relative expression of TNFα (NAM LD: −44%; NAM HD: −57%). An improved anti-inflammatory pattern was reproduced in macrophages cultured in the presence of NAM. Thus, dietary NAM supplementation in ApoE-deficient mice prevented the development of atherosclerosis and improved protection against ApoB-containing lipoprotein oxidation and aortic inflammation.

## 1. Introduction

Coronary artery disease (CAD) represents one of the main outcomes of cardiovascular disease [1]. Atherosclerosis is an inflammatory process that is characterized by the infiltration of macrophages and other inflammatory cell subsets in the arterial wall and subsequently contributes to cardiovascular disease [2,3,4]. Statins have been proven to be effective at reducing cardiovascular disease-related mortality and morbidity [5]. However, a substantial risk of adverse cardiovascular outcomes persists [6,7]. Therefore, over the last few decades, researchers have expressed increasing interest in investigating other pharmacological agents to further decrease the residual cardiovascular disease risk [8].

Nicotinic acid (NA) administration produces favorable antiatherogenic effects in vivo [9,10,11,12,13,14,15,16,17,18] and on humans [19]. NA significantly reduced CAD progression or clinical cardiovascular events in several studies [20]. However, the results from these earlier clinical trials have not been replicated in more recent trials that also included statins [20]. The occurrence of adverse side effects of NA therapy has also limited its use in clinical practice.

Dietary supplementation with other vitamin B_3_ derivatives, such as nicotinamide mononucleotide (NMN) and nicotinamide riboside (NR), confer favorable age-related vasoprotective effects by reversing endothelial dysfunction, attenuating oxidative stress, or rescuing age-related changes in gene expression [21,22,23,24,25]. However, these favorable effects have not been linked to changes in the inflammatory status of target tissues in these studies. Moreover, atherosclerosis was not directly assessed in these studies.

Nicotinamide (NAM), the amide form of vitamin B_3_, has also been shown to exert potent anti-inflammatory effects on chronic inflammatory diseases, including intestinal [26] or dermatological diseases [27,28,29,30]. Consistent with these findings, a dietary deficiency of vitamin B_3_ forms has been associated with different inflammatory processes, including dermatitis, irritation, inflammation of mucous membranes and tubular inflammation [31,32,33,34,35]. The anti-inflammatory effects of NAM have also been explored in vitro in immune cell types [36,37,38,39,40,41,42,43,44,45], which are involved in chronic inflammatory processes such as atherosclerosis. Notably, in one of these studies [35], the authors proposed that NAM may promote monocyte differentiation into macrophages with restricted inflammatory traits.

Increased oxidation often underlies inflammation and triggers cardiovascular disease [46]. Importantly, NAM also exerts a beneficial effect on oxidative stress [47]. Indeed, this vitamin B_3_ form prevents both protein and lipid oxidation [48], at least in part through its ability to scavenge reactive oxidative species [47]. Antioxidant effects of other vitamin B_3_ derivatives, such as NR [49] and NMN [18], on vascular cells have also been reported.

Based on experimental evidence, NAM might also protect against atherogenesis in vitro [36,37,38,39,40,41,42,43,44,45]; however, the effect of NAM has not yet been assessed in vivo. We therefore tested the hypothesis that the administration of NAM in vivo prevents the development of atherosclerosis in a murine model of substantial hypercholesterolemia and atherosclerosis.

## 2. Materials and Methods

All animal procedures were reviewed and approved by the Institutional Animal Care and Use Committee of the Institut de Recerca de l’Hospital de la Santa Creu i Sant Pau (Procedure ID 10434). The methods were conducted in accordance with the approved guidelines. The effect of NAM supplementation on male apolipoprotein (Apo)E-deficient mice on a C57BL/6J genetic background was examined. Two doses of NAM (high dose: 1%; low dose: 0.25%) (cat#N0636, Merck KGaA, Darmstadt, Germany) were administered to mice in the drinking water, starting at the same time as the high-fat diet was initiated. The design of the intervention is shown in Appendix A.

The biochemical parameters were analyzed using commercial kits and adapted to a COBAS 6000/c501 autoanalyzer (Roche Diagnostics, Basel, Switzerland), as described in a previous study [50]. NAM and me-NAM levels were analyzed using high-performance liquid chromatography (HPLC) with mass spectrometry (MS). Plasma concentrations of cytokines (IL-10, IL-6, IL-4 and TNFα) were analyzed using Luminex xMAP^®^ technology (Millipore Corporation, Billerica, MA, USA). The susceptibility of mouse lipoproteins to copper-induced lipid oxidation and the capacity of NAM to inhibit the oxidative modification of human low-density lipoproteins (LDL) were measured by monitoring the formation of conjugated dienes at a λ of 234 nm at 37 °C for 6 h using a BioTek Synergy HT spectrophotometer (BioTek Synergy, Winooski, VT, USA) [51]. Serum oxidized LDL (OxLDL) levels were determined using a murine OxLDL sandwich ELISA (cat# SEA527Mu-96T, Cloud-Clone Corp, Houston, TX, USA). Cellular cholesterol efflux induced by human high-density lipoprotein (HDL) was determined in vitro using [^3^H]-cholesterol-labeled J774A.1 mouse macrophages (ATCC^®^ TIB67™, Manassas, VA, USA), as previously described [52]. Radioactivity counts were measured in both the cell culture medium and cell extract, and the percentage of cholesterol efflux was calculated.

Proximal aortic atherosclerotic lesions in mice were evaluated in serial cryosections of optimal cutting temperature (OCT) compound-embedded preparations. Cryosections were stained with Oil Red O for lipids and the lesion area (surface area stained with Oil Red O) was quantified using AxioVision image analysis software (Zeiss, Jena, Germany). For immunohistochemical staining, thoracic aorta segments were fixed with a 10% neutral buffered formalin solution (cat# HT501128, Merck KGaA, Darmstadt, Germany). Seven-micrometer sections of paraffin-embedded tissue samples were incubated with mouse monoclonal antibodies against IL-10 (cat# GTX632359, diluted 1:50, v:v) and rabbit polyclonal antibodies against TNFα (cat# GTX110520, diluted 1:200, v:v) from GeneTex and stained with diaminobenzidine (DAB) in a Dako Autostainer Link 48 using the Dako EnVision+System-HRP-DAB-kit, according to the manufacturer’s protocol. Slides were then dehydrated and coverslipped and images were obtained using a BX61 Olympus bright-field microscope. Images were quantified using ImageJ-Fiji software. RNA was extracted from tissues and reverse transcribed, and the relative mRNA levels of each gene were determined using fluorescence-based quantitative real time PCR (qPCR) (Appendix A).

The data are presented as medians (interquartile ranges). Statistical analyses were performed using GraphPad Prism software (GPAD, version 5.0, San Diego, CA, USA). The effects of NAM administration on gross and biochemical parameters, histological traits, and gene expression levels were determined using a nonparametric Kruskal–Wallis test followed by the Dunn multiple comparison test. Spearman’s rho correlation coefficients were calculated to determine the correlations between atherosclerosis and parameters of lipoprotein function, with all parameters considered as nonparametric variables. Differences between groups were considered statistically significant when the *p* value was <0.05. Additional details about the methods are available in the online version of the paper (Appendix A).

## 3. Results

### 3.1. Effect of NAM on Gross Parameters and Systemic Phenotype

Plasma NAM concentrations were increased in a dose-dependent manner (NAM low-dose (LD): ~48-fold, *p* < 0.05; NAM high-dose (HD): ~145-fold, *p* < 0.05) in NAM-treated mice (Table 1). The plasma level of the methylated form of NAM (me-NAM) was also increased (~6-fold, *p* < 0.05) in mice receiving the maximal dose. Body weight gain was significantly reduced only in mice receiving the highest dose of NAM (*p* < 0.05) (Table 1). The latter was not accompanied by changes in daily food intake (Table 1). Supplementation with NAM did not affect the liver size (Table 1). Plasma levels of alanine aminotransferase were reduced (NAM LD: 40%, NAM HD: 56%; *p* < 0.05) in NAM-treated mice compared to untreated mice, while plasma aspartate aminotransferase levels did not differ among groups. Plasma creatinine concentrations did not change with NAM treatment (Table 1). Plasma glucose and insulin levels in NAM HD-treated mice did not significantly differ from the levels in untreated mice. However, total cholesterol levels were significantly increased in NAM HD-treated mice (~1.8-fold, *p* ˂ 0.05) compared to untreated mice, exclusively due to increased levels of the non-HDL cholesterol fraction (Table 1). Notably, plasma cholesterol and non-HDL cholesterol levels were approximately 0.7-fold lower in NAM LD-treated mice than in NAM HD-treated mice, which should be considered in the interpretation of the analysis of atherosclerosis.

### 3.2. NAM Administration Prevents the Development of Aortic Atherosclerosis

NAM supplementation significantly decreased the areas of aortic atherosclerotic lesions in treated male mice (NAM LD: 0.55-fold, −45%, *p* ˂ 0.05; NAM HD: 0.45-fold, −55%, *p* ˂ 0.05) compared with untreated mice (Figure 1 and Appendix A). Representative images of the lesions observed in each group are shown. Mice treated with NAM developed less advanced atherosclerotic lesions that were mainly restricted to aortic valves compared with the larger lesions that extended to the free aortic wall in untreated mice. The areas of aortic atherosclerotic lesions were also significantly reduced in female NAM-treated mice (NAM LD: 0.6-fold, *p* ˂ 0.05; NAM HD: 0.3-fold, *p* ˂ 0.05) compared with untreated mice (Appendix A).

The area of atherosclerotic plaques was inversely correlated with the plasma NAM concentration (Spearman’s r = −0.45, *p* ˂ 0.05, *n* = 24).

### 3.3. NAM Administration Directly Protects against the Oxidation of Non-HDL Lipoproteins

The oxidative modification of LDL leads to cholesterol accumulation in macrophages and foam cell formation during atherosclerosis [53]. Notably, the susceptibility of non-HDL particles to oxidation, as measured by conjugated diene formation, was significantly delayed (~2-fold, *p* < 0.05) in NAM HD-treated mice (Figure 2, panels (a) and (b)).

Serum oxLDL concentrations were lower (~35%, *p* < 0.05) in the NAM HD mice than in untreated mice, concomitant with the elevated levels of non-HDL cholesterol in these mice (Figure 2, panel (c)). However, serum oxLDL levels were not associated with the area of atherosclerotic lesions (Appendix A).

The LDL fraction was isolated, and the effect of NAM was analyzed in vitro to determine whether NAM directly prevented LDL oxidation. NAM protected human LDL from oxidation (Figure 2, panel (d)), as observed by a dose-dependent decrease in the rate of conjugated diene production.

### 3.4. The NAM Treatment Improves Plasma and Aortic Inflammation

Plasma concentrations of the anti-inflammatory cytokine IL-10 were significantly increased (NAM HD: ~4-fold, *p* ˂ 0.05) in NAM HD-treated mice compared with untreated mice (Figure 3, panel (a)), whereas the circulating IL-4, TNFα and IL-6 levels did not differ significantly among groups.

The gene expression analysis revealed a significant increase (two-fold, *p* < 0.05) in the relative levels of the *Il10* mRNA in the aorta of NAM HD-treated mice compared with untreated mice (Figure 3, panel (b)). However, the levels of the *Tnfa* mRNA were unchanged. The expression of the *Adgre1* mRNA, which encodes the F4/80 macrophage marker, did not differ among groups, suggesting that macrophage infiltration was not altered by NAM. Both the plasma concentration of IL-10 and relative levels of the *Il10* mRNA in the aorta were inversely correlated with atherosclerosis in ApoE-deficient mice (plasma IL-10 concentration: Spearman’s r = −0.46, *p* < 0.05; aortic expression of the *Il10* mRNA: Spearman’s r = −0.56, *p* < 0.05) (Figure 3, panels (c) and (d)). Immunohistochemical staining of thoracoabdominal aortas showed an increase in the relative abundance of IL-10 in both NAM LD-(~1.3-fold, *p* < 0.05) and NAM HD-treated mice (~1.2-fold, *p* < 0.05) compared with untreated mice (Figure 3, panel (e)). Importantly, the relative abundance of TNFα in thoracoabdominal aortas was significantly decreased in both groups of treated mice (NAM LD: ~0.7-fold, *p* < 0.05; NAM HD: ~0.5-fold, *p* < 0.05) compared with untreated mice (Figure 3, panel (e)). Negative controls further validated the results of immunohistochemical staining in aortic tissue (Appendix A). These observations suggest the existence of posttranscriptional mechanisms that modulate the levels of IL-10 and TNFα in the aorta. Only the relative thoracoabdominal aortic level of IL-10 showed a marginal trend towards an inverse correlation with atherosclerosis in ApoE-deficient mice (plasma IL-10 level: Spearman’s r = −0.47, *p* = 0.06) (Appendix A). The expression of the *Tnfa*, *Il6* and *Il1b* mRNAs was significantly reduced in lipopolysaccharide (LPS)-activated J774A.1 macrophages incubated with different NAM doses in a dose-dependent manner (Figure 3, panel (f)). Unfortunately, the levels of the *Il10* mRNA were undetectable in these cells.

As a precursor of NAD+ [54], NAM conceivably increases sirtuin (SIRT)1 activity in aortic tissue. Notably, SIRT1 induces liver X receptor (LXR) function [55], which in turn controls the expression of key transporters, i.e., *Abca1* and *abcg1* in macrophages [56,57], involved in cholesterol efflux. Thus, the relative mRNA levels of the abovementioned LXR targets were directly determined in thoracoabdominal aortas and cultured macrophages (Appendix A). *Abca1* was upregulated in NAM LD-(~2-fold, *P* ˂ 0.05) and NAM HD-treated mice (~4-fold, *p* ˂ 0.05), whereas *Abcg1* expression was marginally increased in NAM HD-treated mice (2-fold, *p* = 0.07) (Appendix A). Interestingly, the NAM HD treatment induced *Nr1h2* expression (encoding LXRβ) (1.7-fold, *p* ˂ 0.05) without changing the expression of *Nr1h3* (encoding LXRα) (Appendix A). Only the relative levels of the *Abca1* mRNA were inversely correlated with aortic atherosclerosis (*Abca1* mRNA: Spearman’s r = −0.42, *p* < 0.05), while relative levels of the *Abcg1* mRNA only showed a nonsignificant trend towards an association with the area of atherosclerotic lesions (Spearman’s r = −0.40, *p* = 0.06) (Appendix A). Similarly, the *Abca1* mRNA was upregulated in cultured macrophages exposed to NAM (Appendix A); only a marginal effect was observed on *Abcg1* expression. Moreover, the cholesterol efflux capacity induced by a common source of human HDL showed a moderate, but not significant, increasing trend (*p* = 0.07) in NAM-treated cells (at the highest concentration assayed, 10 mM) compared with untreated cells (Appendix A).

## 4. Discussion

Based on our data, the administration of NAM prevented atherosclerosis and inflammation, despite the substantial concomitant increase in plasma non-HDL-cholesterol levels. Inflammation is frequently characterized by increased plasma concentrations of a number of pro-inflammatory markers (e.g., IL-6 and TNFα) and decreased levels of anti-inflammatory cytokines, such as IL-10 [38]. The hypothesis that NAM is an anti-inflammatory factor has also been supported by different studies [36,37,38,39,40,41,42,43,44,45]. As shown in the present study, plasma concentrations of IL-10 were significantly increased in NAM-treated ApoE-deficient mice. Notably, the relative concentrations of the IL-10 and TNFα proteins were significantly altered in the aortas of NAM-treated mice. This finding is consistent with previous data [39] showing an effect of NAM on inhibiting *Tnfa* synthesis and secretion in vitro. As TNFα is mainly released by activated M1 macrophages [58], NAM might predominantly exert its anti-inflammatory effect by reducing TNFα synthesis and increasing IL-10 production in resident macrophages in the aorta. Although the infiltration and activation of immune cells is a characteristic of chronic inflammation processes, such as atherosclerosis [59], the analyses of different molecular surrogates of macrophage infiltration, i.e., *Adgre1* and *Cd68*, in the present study did not provide evidence suggesting the differential accumulation of macrophages in the aortas of ApoE-deficient mice.

Oxidative stress is usually regarded as a pro-inflammatory condition [60]. Conceivably, the anti-inflammatory effect of NAM may also involve antioxidant mechanisms. Indeed, NAM has been defined as an O· radical scavenger and may also inhibit free radical (e.g., NO·, O·, and HCLO·) generation [47], protecting against both protein oxidation and lipid peroxidation induced by reactive oxygen species [46]. Although non-HDL cholesterol accumulated in the plasma of NAM HD-treated, ApoE-deficient mice, mainly due to impaired plasma clearance [61], these lipoproteins were less susceptible to oxidation than the lipoproteins from untreated mice. This direct antioxidant effect of NAM might contribute to preventing atherosclerosis development, despite the increase in the plasma non-HDL level. As previously reported [59], the mechanistic basis for the hypercholesterolemic effect of NAM HD is due to a delayed clearance of non-HDL lipoproteins that appears to depend strongly on the lack of ApoE, as it has not been observed in wildtype mice (Méndez-Lara et al. unpublished data) treated with a similar NAM dosage.

Unlike NAM, the administration of other dietary vitamin B_3_-related metabolites (i.e., NMN and NR) has previously been reported to protect the vasculature from oxidative stress [22,23]. As NMN and NR are NAD+ intermediaries [54], this effect might be at least partially attributed to SIRT1 signaling in target tissues [23]. NAM is also a NAD+ precursor [54]; therefore, its administration might provide tissues with an extra source of NAD+ and induce SIRT1 activity in the aortic tissues of treated mice. Although many studies have been designed based on the premise that NAM functions as a potent inhibitor of SIRT1 in vitro, compelling evidence now suggests that NAM may exert the opposite effect in vivo [62]. For instance, NAM, by virtue of its role as an NAD+ precursor [54], would potentially drive SIRT1 activity by increasing cellular NAD + pools [63]. Some key ABC transporters (i.e., *Abca1* and *Abcg1*) involved in the first step of reverse cholesterol transport, cholesterol efflux [64], were positively upregulated in the aortas of NAM-treated mice (Appendix A). Interestingly, *Abca1* and *Abcg1* expression are controlled by the nuclear receptor LXR, suggesting that this signaling pathway might be induced in NAM HD-treated mice [55]. The observation that only the highest dose of NAM exerted the main effects on gene expression suggests that NAM functions as a weak LXR activator. Moreover, the levels of the *Nr1h2* mRNA, which encodes LXRβ, were significantly increased in aortas from NAM HD-treated mice (*Nr1h2:* two-fold, *p* ˂ 0.05) compared with untreated mice. LXRα, which is encoded by the *Nr1h3* gene, may regulate its own expression in human macrophages [65,66], but not the expression of *Nr1h2* [65]. LXRβ activation is sufficient to reduce atherosclerosis [67] and may contribute [68], together with LXRα, to the favorable upregulation of *Abca1* and *Abcg1* expression both in vivo and in vitro.

Cholesterol efflux from plaque macrophages is an important process contributing to the removal of excess cholesterol from the artery wall. However, cholesterol efflux is only the first step of the overall process, and, importantly, macrophage-specific reverse cholesterol transport to feces in vivo (m-RCT) was impaired in NAM HD-treated ApoE-deficient mice [61]. Therefore, delayed plasma clearance of non-HDL might underlie defective m-RCT [61], as observed in LDL receptor knockout mice [69]. Overall, the detrimental effect of NAM HD treatment on m-RCT, in the context of a concomitantly severely worsening hyperlipidemia, highlights the antiatherogenic power of this compound at least in mice. Our data reveal a common NAM-related change in the gene expression pattern of cholesterol transporters and anti-inflammatory cytokines in the aortas of treated mice and macrophages. Accumulating evidence supports the hypothesis of molecular crosstalk between the cholesterol transporters ABCA1/ABCG1 and the immune system that will provide a greater benefit in terms of alleviating inflammation than m-RCT in this case [70,71,72,73,74,75].

### Limitations of the Study

The present study has several limitations. First, most of the experiments described in this study were only performed in male mice. However, we also provided evidence of decreased atherosclerosis in NAM HD-treated female mice. Second, some of the observed changes in mRNA levels without an examination of protein levels or functions do not necessarily reflect changes in the protein content and activity. Third, plasma NAM concentrations were not determined at the beginning of the experiment; however, genetically identical untreated ApoE-deficient mice had significantly lower plasma concentrations of NAM than NAM-treated mice. Fourth, me-NAM, a metabolic product of NAM [54], is also atheroprotective [76,77]. Thus, by increasing the plasma concentration of me-NAM, NAM administration might also contribute to preventing atherosclerosis development in vivo. Although we were unable to exclude a beneficial effect of me-NAM, its individual effect was not directly assessed in the present work. Fifth, the potential involvement of a SIRT1-mediated mechanism was only indirectly revealed by the increased expression of some target LXR genes in aortas of NAM HD-treated mice. Finally, the experimental design used in this study assessed the preventive but not the therapeutic effects of NAM on atherosclerosis. Thus, further studies are warranted to confirm and extend the present observations.

## 5. Conclusions

NAM supplementation prevents the formation of aortic lesions in ApoE-deficient mice in a dose-dependent manner, which is related to increases in the circulating and relative aortic levels of the IL-10 mRNA and protein, as well as reductions in the level of the TNFα protein in thoracoabdominal aortas, suggesting a switch towards anti-inflammatory macrophages. The susceptibility of non-HDL to oxidation was improved by NAM in vitro and in vivo, thus suggesting another mechanism by which NAM protects against the development of atherosclerosis in NAM-treated mice. The moderate induction of the expression of key cholesterol transporters involved in cholesterol removal in aortas of treated mice might reflect molecular crosstalk, although these changes would not result in increased m-RCT in this mouse model.

## Figures and Tables

**Figure 1 antioxidants-09-01162-f001:**
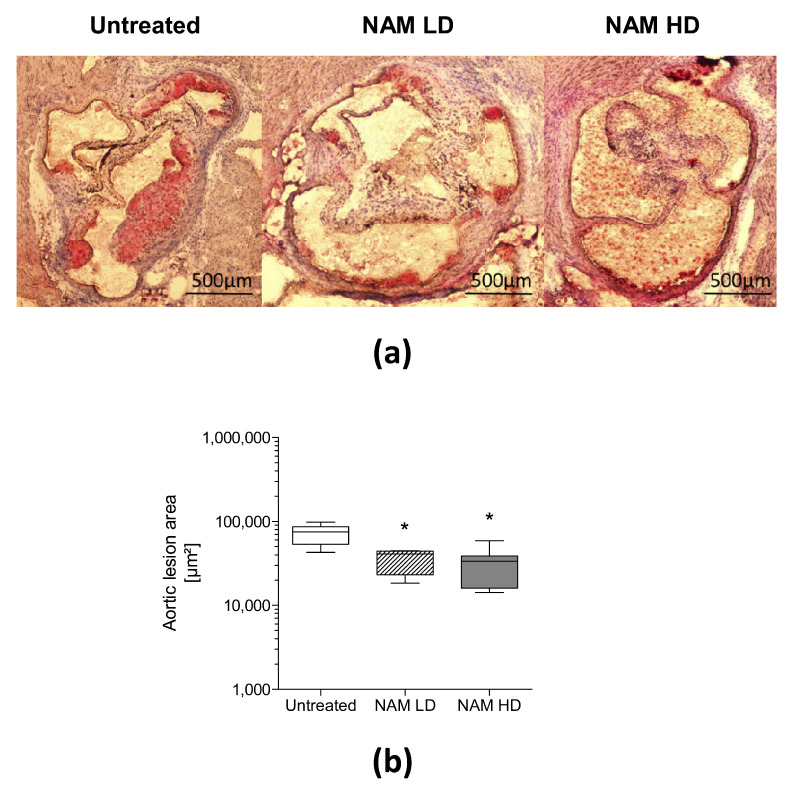
Analysis of proximal aortic lesions in apolipoprotein (Apo)E-deficient mice treated with different doses of NAM. (**a**) Representative images of aortic atherosclerotic lesion in 12-week-old mice challenged to a western diet and NAM over 1 month at 2 months of age. (**b**) Area of proximal aortic lesion quantified from 8 mice per group. Data are expressed as the median (interquartile range) of four consecutive sections throughout the aortic sinus that were obtained every 20 µm when aortic valves became visible. Statistically significant differences among groups for each variable were determined using a nonparametric a Kruskal–Wallis test followed by Dunn’s posttest. Differences were considered significant when *p* < 0.05. Specifically, * *p* < 0.05 vs. Untreated group. Abbreviations used: NAM LD, low-dose, NAM-treated mice; NAM HD, high-dose, NAM-treated mice.

**Figure 2 antioxidants-09-01162-f002:**
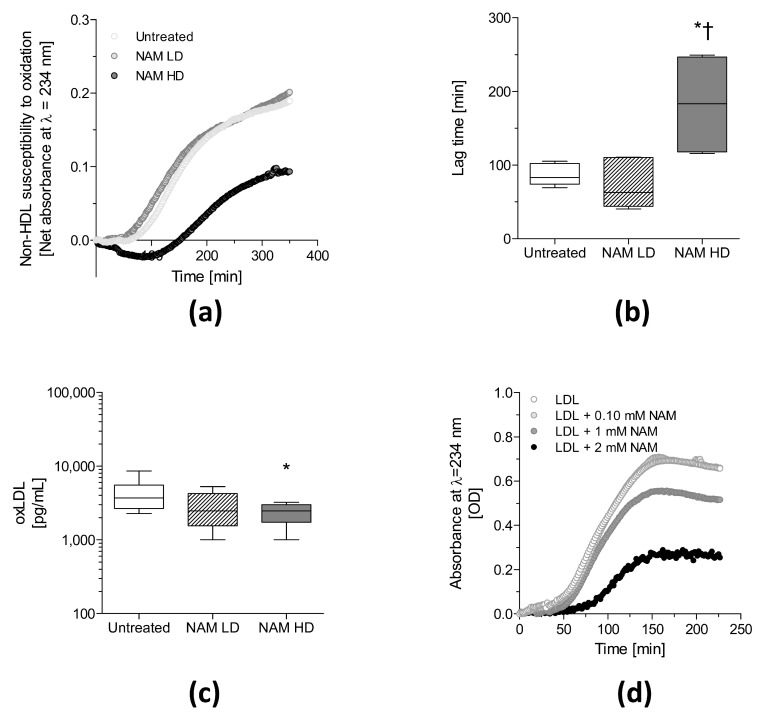
Antioxidant effects of NAM on ApoE-deficient mice. Two-month-old male ApoE-deficient mice were challenged with a Western diet and NAM for 1 month. (**a**) Representative diene formation curves of non-HDL particles. (**b**) Median diene formation lag time calculated from the oxidation curves (*n* = 3–4 plasma pools/group). (**c**) Serum oxLDL concentration (.*n* = 8 per group). (**d**) Oxidation curves of human LDL incubated at NAM concentration 0.10 mM and 1.0 mM. This experiment was replicated twice with similar outcomes. Oxidation kinetics were carried using plasma lipoproteins isolated by sequential ultracentrifugation from pools of 2–3 mouse plasmas of each experimental group, panels (**a**,**b**), or pooled human plasma, panel (**d**), as appropriate. In panels (**b**,**c**), data are expressed as medians (interquartile ranges). Statistically significant differences among groups for each variable were determined using a nonparametric a Kruskal–Wallis test followed by Dunn’s posttest. Differences were considered significant when *p* < 0.05. Specifically, * *p* < 0.05 vs. Untreated group or † *p* < 0.05 vs. NAM LD-treated mice. Abbreviations used: NAM LD, low-dose, NAM-treated mice; NAM HD, high-dose, NAM-treated mice; OD, optical density.

**Figure 3 antioxidants-09-01162-f003:**
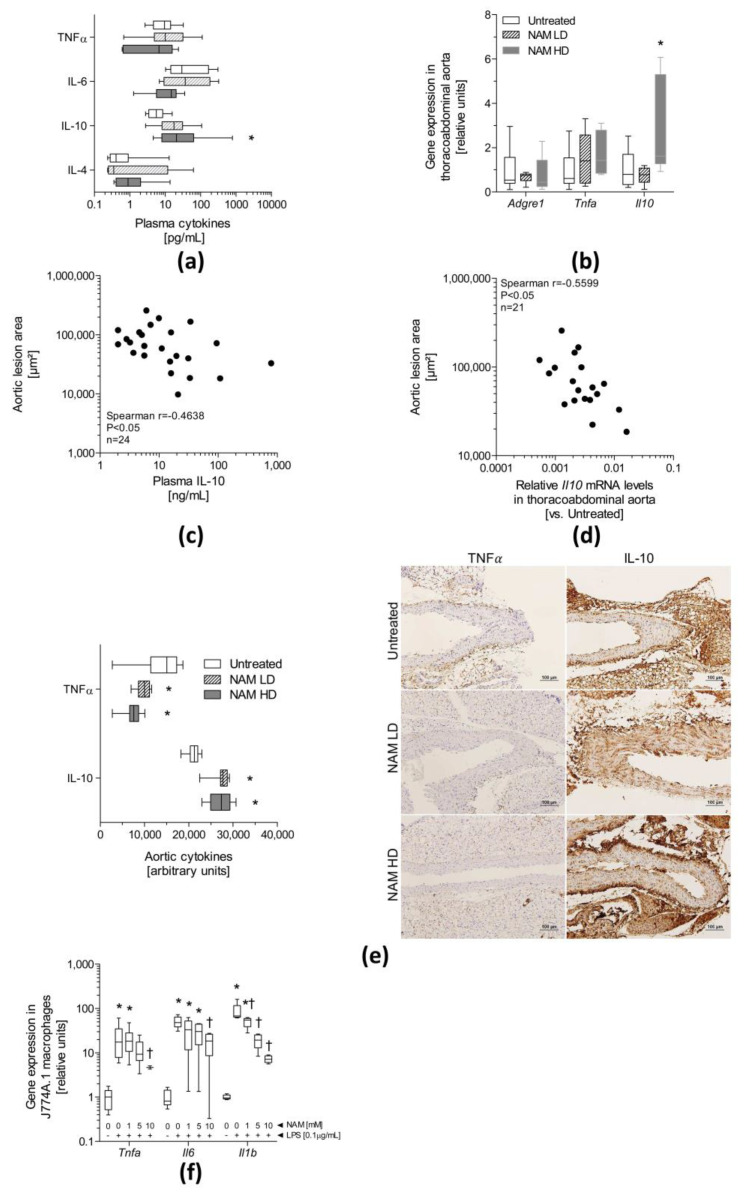
Anti-inflammatory effects of NAM on ApoE-deficient mice and cultured macrophages. Two-month-old male ApoE-deficient mice were challenged with a Western diet and NAM for 1 month. (**a**) Plasma concentration of cytokines (TNFα, IL-6, IL-10, and IL-4) in ApoE-deficient mice (*n* = 5–6). (**b**) Relative aortic mRNA levels of inflammatory targets. (**c**) Correlation between the area of proximal aortic lesions and circulating IL-10 levels. (**d**) Correlation between the area of proximal aortic lesions and *Il10* aortic mRNA levels. (**e**) Immunohistochemical (IHC) analysis of TNFα and IL-10 levels in the aorta. Left panel, bar chart showing the relative (arbitrary units) expression of TNFα and IL-10 in the aortas from different groups (*n* = 5–6 mice per group). Right panel, representative images of immunohistochemical staining for IL-10 and TNFα in thoracoabdominal aortas. (**f**) Relative mRNA levels of cytokines assessed in LPS-activated J774A.1 macrophages exposed to different doses of NAM for 24 h. Data are expressed as the median (interquartile range) of 4 independent experiments. In panels (**a**,**b**,**e**), data are presented as medians (interquartile ranges) (*n* = 5–6 samples/group) and from 4 independent experiments in panel F. Statistically significant differences among groups for each variable were determined using a nonparametric a Kruskal–Wallis test followed by Dunn’s posttest. Differences were considered significant when *p* < 0.05. Specifically, * *p* < 0.05 vs. Untreated group or † *p* < 0.05 vs. NAM HD-treated mice. In panels (**c**,**d**), the relationship between parameters was tested using a nonparametric Spearman’s correlation test. Mice of all groups were considered for analysis. Abbreviations used: LPS, Lipopolysaccharide; NAM, nicotinamide; NAM LD, low-dose, NAM-treated mice; NAM HD, high-dose, NAM-treated mice.

**Table 1 antioxidants-09-01162-t001:** Effect of NAM administration on gross parameters and plasma biochemical parameters of male ApoE-deficient mice.

Parameters	Untreated	NAM LD	NAM HD	*p*
*Gross parameters*				
Body weight (g)	28.7 (27.0; 29.9)	27.4 (23.3; 28.1)	24.6 (22.3; 26.0) *	<0.05
Liver weight (g)	1.3 (1.2; 1.4)	1.3 (1.2;1.4)	1.2 (1.1;1.4)	0.20
Diet intake (g/day)	2.5 (2.4; 2.8)	2.7 (2.6; 2.8)	2.4 (2.1; 2.5) †	<0.05
Water intake (g/day)	3.9 (3.2; 5.1)	4.6 (4.4; 4.8)	4.6 (3.8; 5.4)	0.20
Calculated dose of NAM (g/kg/day)	-	0.5 (0.4; 0.7)	1.9 (1.6; 2.2) †	<0.05
*Plasma biochemistry*				
NAM (µM)	4.0 (3.3; 5.5)	193.5 (90.5, 248.5)	580.0 (526.0; 605.0) *	<0.05
me-NAM (relative values) (×10^−3^) ^a^	0.15 (0.13; 0.16)	0.28 (0.19; 0.31)	0.85 (0.45; 0.89) *	<0.05
Glucose (mM)	12.3 (10.6; 15.3)	11.1 (9.4; 11.8)	10.9 (8.4; 12.9)	0.17
Insulin (μg/L)	0.7 (0.6; 0.8)	n. d.	0.7 (0.6; 0.7)	0.32
Triglycerides (mM)	0.5 (0.4; 1.0)	1.8 (1.3; 2.5) *	1.5 (1.1; 2.3) *	<0.05
Total cholesterol (mM)	43.3 (33.7; 47.4)	53.0 (44.2; 55.2)	77.6 (71.6, 82.3) * †	<0.05
Non-HDL cholesterol (mM)	43.1 (33.7; 47.0)	52.6 (43.8; 54.9)	77.3 (71.2, 82.0) * †	<0.05
HDL cholesterol (mM)	0.2 (0.1; 0.4)	0.4 (0.3; 0.4)	0.4 (0.1; 0.5)	0.45
AST (U/L)	61 (24;128)	25 (19; 39)	63 (27; 117)	0.33
ALT (U/L)	16 (7; 48)	5 (3; 6) *	6 (2; 11) *	<0.05
Creatinine (mM)	0.02 (0.01; 0.03)	0.01 (0.01; 0.02)	0.01 (0.01; 0.02)	0.16

Results are reported as medians (interquartile ranges) (*n* = 8 mice per group). All analyses were conducted in three-month-old mice. At two months of age, the mice were challenged with a Western diet, and treated with or without NAM for 1 month. Food and water intake was measured at the end of the study as described in the Materials and Methods section (Appendix A). Plasma concentration of NAM was expressed in µM, whereas that of me-NAM were shown in relative values (*n* = 5–6 mice per group). Plasma levels of the HDL fractions were determined in the plasma supernatants after precipitating with phosphotungstic acid (Roche); the non-HDL fraction was calculated by subtracting the HDL moiety from the total plasma. Statistically significant differences among groups for each variable were determined using a nonparametric a Kruskal–Wallis test followed by Dunn’s posttest; differences were considered significant when *p* < 0.05. Specifically, * *p* < 0.05 vs. Untreated group; † *p* < 0.05 vs. NAM LD group. Abbreviations used: ALT, alanine aminotransferase; AST, aspartate aminotransferase; NAM, nicotinamide, me-NAM, methylated form of nicotinamide; NAM LD, low-dose, NAM-treated mice; NAM HD, high-dose, NAM-treated mice, HDL, high-density lipoprotein; n. d., not determined. ^a^ relative units.

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
