# Peer review of "Nicotinamide Prevents Apolipoprotein B-Containing Lipoprotein Oxidation, Inflammation and Atherosclerosis in Apolipoprotein E-Deficient Mice"

_antioxidants, 2020, doi:10.3390/antiox9111162_

Round 1

Reviewer 1 Report

In this manuscript, Karen A. Mendez-Lara and colleagues provided the effect of NAM on the development of atherosclerosis in a mouse model of the disease. ApoB-containing lipoproteins from high dose of NAM-treated mice were less prone to oxidation, although circulating ApoB-containing lipoproteins increased. In addition, immunohistochemical staining of aortas from NAM-treated mice exhibited increased IL-10 and decreased TNF-alpha.

This work is well performance and the manuscript is carefully written. Nevertheless, there are some issues that needed to be written.

non-HDL cholesterol increased in high dose NAM treated mice in this study. What kind of lipoproteins increased? LDL, VLDL, remnant lipoprotein, HDL?  Did the authors check the lipoprotein distribution using, for example, lipoprotein electrophoresis or HPLC method.

The oxidative modification of LDL plays a key role in the development of atherosclerosis. However, the authors experimented the susceptibility of non-HDL, but not LDL, in mouse study. In contrast, the authors experimented the susceptibility of LDL from human sera. Why did the authors experiment the susceptibility of LDL in mouse study? I think oxidation of LDL, but not non-HDL, is well known to an initiator of atherosclerosis.

Line 93, Please disclose where to purchase the NAM used in the experiment.

Supplementary Fig.2 (a) (d) look like a time series. In fact, the measurements of 4 consecutive sections of the aorta are simply  lined up. Then, Supplementary Fig.2(a) and Fig.1(b) are identical, and Supplementary Fig.2(c) and (d) are identical, I think.

As the authors discussed, NAM may have activated SIRT1. Did the authors check SIRT1 activity or expression?

Line 195, Can the oxLDL fraction be isolated? Isn't it a mistake of LDL instead of oxLDL?

Author Response

Reviewer #1

In this manuscript, Karen A. Mendez-Lara and colleagues provided the effect of NAM on the development of atherosclerosis in a mouse model of the disease. ApoB-containing lipoproteins from high dose of NAM-treated mice were less prone to oxidation, although circulating ApoB-containing lipoproteins increased. In addition, immunohistochemical staining of aortas from NAM-treated mice exhibited increased IL-10 and decreased TNF-alpha.

This work is well performance and the manuscript is carefully written. Nevertheless, there are some issues that needed to be written.

Thank you for your criticisms and comments that helped improve the manuscript.

1- non-HDL cholesterol increased in high dose NAM treated mice in this study. What kind of lipoproteins increased? LDL, VLDL, remnant lipoprotein, HDL?  Did the authors check the lipoprotein distribution using, for example, lipoprotein electrophoresis or HPLC method.

Only the plasma fraction of non-HDL was found elevated in NAM-treated ApoE-deficient mice. Unfortunately, the lipoprotein profile was not further analyzed by either fast protein liquid chromatography nor native gradient gel electrophoresis; however, we previously reported and increased plasma non-HDL cholesterol observed in NAM HD treated ApoE-deficient mice due to defective clearance of these lipoproteins [1]. The lack of apoE, a critical lipoprotein receptor ligand, in these mice leads to the plasma accumulation of non-HDL cholesterol which is mainly accounted for delayed clearance and hence increased concentrations of remnant lipoproteins.  As explained in the ms, NAM HD enhances this effect, an interaction that was not seen in wild type animals in which NAM HD does not increase non-HDL cholesterol.

2- The oxidative modification of LDL plays a key role in the development of atherosclerosis. However, the authors experimented the susceptibility of non-HDL, but not LDL, in mouse study. In contrast, the authors experimented the susceptibility of LDL from human sera. Why did the authors experiment the susceptibility of LDL in mouse study? I think oxidation of LDL, but not non-HDL, is well known to an initiator of atherosclerosis.

The term non-HDL cholesterol in mice would include VLDL cholesterol, IDL (remnant)-cholesterol and LDL-cholesterol. It is worth noting that, in contrast to human, these lipoprotein classes contain a mixture of apoB-48 and apoB-100, being usually LDL with apoB-100 a quantitatively minor lipoprotein.  This could explain the more intensive use of the term non-HDL cholesterol and, also, that the non-HDL lipoproteins as a whole are subjected to oxidation susceptibility studies. 

LDL oxidative modification is certainly a hallmark of atherosclerosis formation [2-6] but, noteworthy, elevated lipoprotein remnants also contribute to atherogenesis in humans [7, 8] and mice [9]. Importantly, our data clearly showed that NAM manipulation exerts a favorable antioxidant action on atherogenic lipoproteins both in vivo and ex vivo. Thus, data from both experiments showed a consistent protective effect of NAM on plasma ApoB-containing lipoproteins and, further a second experiment with isolated human LDL showed that the favorable effect was directly due to NAM. Consistent with these findings in susceptibility to oxidation, plasma levels of oxLDL was concomitantly reduced in mice treated with NAM HD.

Lastly, one of the potential antiatherogenic actions of HDL is to protect LDL from oxidation. This is the reason why we also measured this capacity.

3- Line 93, Please disclose where to purchase the NAM used in the experiment.

Done.

4- Supplementary Fig.2 (a) (d) look like a time series. In fact, the measurements of 4 consecutive sections of the aorta are simply  lined up. Then, Supplementary Fig.2(a) and Fig.1(b) are identical, and Supplementary Fig.2(c) and (d) are identical, I think.

Actually, the mean lesion area of each of the four consecutive sections are shown in Supplementary Fig. 2(a), whereas a single mean area of the four sections was shown in Fig.1(b). The former gives supplemental spatial information of the atherosclerotic lesion. Suplementary Fig.2(c) and (d) shows also both types of information, but in this case in reference to female mice only.

5- As the authors discussed, NAM may have activated SIRT1. Did the authors check SIRT1 activity or expression?

The direct analysis of SIRT1 activity in aortic tissue could not be eventually determined. A SIRT1-mediated mechanism could so far only be indirectly revealed by virtue of the increased expression of some target LXR genes in aortas of NAM treated mice. The latter was properly acknowledged as a one of the main limitations of the study (lines 348-350) and is therefore deserved to be studied in future studies.

6- Line 195, Can the oxLDL fraction be isolated? Isn't it a mistake of LDL instead of oxLDL?

The mention oxLDL was mistakenly noted; we were referring to the LDL fraction. This has been corrected in the revised version of the ms.

Additional references

1. Mendez-Lara, K. A., Santos, D., Farre, N., Nan, M. N., Pallares, V., Perez-Perez, A., Alonso, N., Escola-Gil, J. C., Blanco-Vaca, F. and Julve, J. (2019) Vitamin B3 impairs reverse cholesterol transport in Apolipoprotein E-deficient mice. Clin Investig Arterioscler. 31, 251-260

2. Cristofori, P., Crivellente, F., Campagnola, M., Pasini, A. F., Garbin, U., Rigoni, A., Tosetti, M., Turton, J., Faustinelli, I. and Cominacini, L. (2004) Reduced progression of atherosclerosis in apolipoprotein E-deficient mice treated with lacidipine is associated with a decreased susceptibility of low-density lipoprotein to oxidation. Int J Exp Pathol. 85, 105-114

3. Aoki, T., Abe, T., Yamada, E., Matsuto, T. and Okada, M. (2012) Increased LDL susceptibility to oxidation accelerates future carotid artery atherosclerosis. Lipids Health Dis. 11, 4

4. Dean, R. T. (2000) Beyond Schuh: early studies on the oxidation of LDL and other lipoproteins and its role in atherosclerosis. Redox Rep. 5, 251-255

5. van de Vijver, L. P., Kardinaal, A. F., van Duyvenvoorde, W., Kruijssen, D. A., Grobbee, D. E., van Poppel, G. and Princen, H. M. (1999) Oxidation of LDL and extent of peripheral atherosclerosis. Free Radic Res. 31, 129-139

6. van de Vijver, L. P., Kardinaal, A. F., van Duyvenvoorde, W., Kruijssen, D. A., Grobbee, D. E., van Poppel, G. and Princen, H. M. (1998) LDL oxidation and extent of coronary atherosclerosis. Arterioscler Thromb Vasc Biol. 18, 193-199

7. Hoeke, G., Wang, Y., van Dam, A. D., Mol, I. M., Gart, E., Klop, H. G., van den Berg, S. M., Pieterman, E. H., Princen, H. M. G., Groen, A. K., Rensen, P. C. N., Berbee, J. F. P. and Boon, M. R. (2017) Atorvastatin accelerates clearance of lipoprotein remnants generated by activated brown fat to further reduce hypercholesterolemia and atherosclerosis. Atherosclerosis. 267, 116-126

8. Takeichi, S., Yukawa, N., Nakajima, Y., Osawa, M., Saito, T., Seto, Y., Nakano, T., Saniabadi, A. R., Adachi, M., Wang, T. and Nakajima, K. (1999) Association of plasma triglyceride-rich lipoprotein remnants with coronary atherosclerosis in cases of sudden cardiac death. Atherosclerosis. 142, 309-315

9. Joven, J., Rull, A., Ferre, N., Escola-Gil, J. C., Marsillach, J., Coll, B., Alonso-Villaverde, C., Aragones, G., Claria, J. and Camps, J. (2007) The results in rodent models of atherosclerosis are not interchangeable: the influence of diet and strain. Atherosclerosis. 195, e85-92

Reviewer 2 Report

A well written and presented study. Clear hypothesis well founded in current stat-of-the-art. Data clearly interrogated, interpreted and presented This study is important and of interest to a wide readership. Only minor points to clarify. 

Ln 59-60 vivo [9-18] and on humans [19]. Please revise

Ln 71 including intestinal [26] or and dermatological diseases [27-30]. Revise

122 using fluorescence-based quantitative real time PCR (qPCR). Direct reader to supplementary M&M

Ln 146-148 Notably, plasma cholesterol and non-HDL cholesterol levels were approximately 0.7-fold lower in NAM LD-treated mice than in NAM HD-147 treated mice, which should be considered in the interpretation of the analysis of atherosclerosis. Please discuss and eloborate

Ln 213 3.4. The NAM treatment improves inflammation. Vague please revise

Author Response

Reviewer #2

A well written and presented study. Clear hypothesis well founded in current stat-of-the-art. Data clearly interrogated, interpreted and presented. This study is important and of interest to a wide readership. Only minor points to clarify.

Thank you for your criticisms and comments that helped improve the manuscript.

1- Ln 59-60 vivo [9-18] and on humans [19]. Please revise

This sentence has been maintained as it was revised by the contracted editing services (American Journal Experts).

2- Ln 71 including intestinal [26] or and dermatological diseases [27-30]. Revise

“and” was a missed typographical error which was not corrected in the previous version of the ms. It has been removed in the revised version of the ms.

3- 122 using fluorescence-based quantitative real time PCR (qPCR). Direct reader to supplementary M&M

Done.

4- Ln 146-148 Notably, plasma cholesterol and non-HDL cholesterol levels were approximately 0.7-fold lower in NAM LD-treated mice than in NAM HD-147 treated mice, which should be considered in the interpretation of the analysis of atherosclerosis. Please discuss and eloborate

A major issue when comparing the effects of the HD and LD NAM treatments is the plasma cholesterol levels of both groups. However, since our data supported the hypothesis that NAM may prevent the development of atherosclerosis mainly by ameliorating the aortic inflammatory status, as revealed by increases in the expression of IL-10 and decreased expression TNFa by immunohistochemical analysis, in treated ApoE-deficient mice, regardless a concomitantly severely worsening hyperlipidemia in NAM-treated mice.

5- Ln 213 3.4. The NAM treatment improves inflammation. Vague please revise

The subtitle “3.4 The NAM treatment improves inflammation” have been replaced by “3.4 The NAM treatment improves plasma and aortic inflammation” in the revised version of the ms
